# Understanding Self-supervised Contrastive Learning through Supervised Objectives

**Byeongchan Lee**  *prinsommer@kaist.ac.kr*
*KAIST*

**Reviewed on OpenReview:** *https://openreview.net/forum?id=cmE97KX2XM*

## Abstract

Self-supervised representation learning has achieved impressive empirical success, yet its theoretical understanding remains limited. In this work, we provide a theoretical perspective by formulating self-supervised representation learning as an approximation to supervised representation learning objectives. Based on this formulation, we derive a loss function closely related to popular contrastive losses such as InfoNCE, offering insight into their underlying principles. Our derivation naturally introduces the concepts of prototype representation bias and a balanced contrastive loss, which help explain and improve the behavior of self-supervised learning algorithms. We further show how components of our theoretical framework correspond to established practices in contrastive learning. Finally, we empirically validate the effect of balancing positive and negative pair interactions. All theoretical proofs are provided in the appendix, and our code is included in the supplementary material.

## 1 Introduction

Representation learning, the process of acquiring condensed but meaningful representations (Bengio et al., 2013; LeCun et al., 2015; Goodfellow et al., 2016), lies at the core of advancing machine learning capabilities. Supervised learning, while effective, depends heavily on labeled data, which can be problematic in the face of diverse and dynamic real-world environments. Human annotation is not only labor-intensive and costly (hard to scale), but also subjective and prone to errors (hard to generalize) (Vasudevan et al., 2022; Beyer et al., 2020; Shankar et al., 2020).

In response to these challenges, self-supervised learning (SSL), motivated by the idea that humans can learn without explicit labels, has shown strong empirical success in domains such as computer vision, natural language processing, and speech recognition (Ozbulak et al., 2023; Schiappa et al., 2023; Gui et al., 2023). While supervised learning is built on well-defined objectives such as empirical risk minimization, self-supervised learning has mainly progressed through architectural innovations, rather than starting from formal objective formulations. Many recent methods adopt a Siamese architecture and combine various techniques such as memory banks, momentum encoders, stop-gradient operations, and multi-view augmentations (Wu et al., 2018; He et al., 2020; Grill et al., 2020; Chen & He, 2021; Caron et al., 2020; 2021; Zbontar et al., 2021; Amrani et al., 2022).

In this paper, we present a theoretical framework that interprets self-supervised representation learning as an approximation of supervised representation learning. While self-supervised representation learning operates without ground-truth labels, it implicitly constructs supervision signals, suggesting an underlying connection to supervised representation learning objectives.[1] To explore this connection, we begin by expressing supervised representation learning as an optimization over similarities to class prototypes. We then approximate this formulation using only unlabeled data and data augmentations, leading to a self-supervised

---

[1]This is implied within expressions such as pseudo labels (Doersch et al., 2015; Noroozi & Favaro, 2016; Zhang et al., 2016; Gidaris et al., 2018), target (or teacher) encoders (Tarvainen & Valpola, 2017; He et al., 2020; Grill et al., 2020; Chen & He, 2021; Caron et al., 2021; Oquab et al., 2023) in the literature.

loss that closely resembles the InfoNCE loss used in SimCLR (Chen et al., 2020a), which serves as a hub for many algorithms. This derivation clarifies how self-supervised representation learning can be understood as solving a surrogate form of supervised representation learning. Additionally, our formulation naturally introduces the concept of *prototype representation bias*, and motivates a *balanced contrastive loss* that improves the approximation. These insights offer a more principled understanding of self-supervised representation learning and its relationship to supervised objectives.

**Contributions** of our work are summarized as follows:

1. We present a theoretical framework that formulates self-supervised representation learning as an approximation of supervised representation learning. From this formulation, we derive a contrastive loss closely related to the InfoNCE loss, providing a principled explanation for its structure.

2. Our framework offers a perspective on common practices in contrastive learning, such as representation normalization and the use of balanced datasets.

3. We introduce the concept of prototype representation bias arising from the approximation, and observe its correlation with downstream performance.

4. We propose a balanced contrastive loss as a natural extension of the InfoNCE loss, and observe that improved balancing leads to better performance.

## 2 Related work

**Contrastive losses** Our work falls into the category of contrastive learning, which is characterized by the use of contrastive losses. The concept of contrastive loss was first introduced in Chopra et al. (2005). Since then, several variants have emerged. The triplet loss simultaneously considers three representations, each serving as an anchor, a positive sample, and a negative sample (Weinberger & Saul, 2009; Chechik et al., 2010). Furthermore, the $(m + 1)$-tuplet loss treats $m + 1$ representations: an anchor, a positive sample, and $m - 1$ negative samples, and it is composed in the form of a softmax function (Sohn, 2016). Wu et al. (2018) combine a temperature parameter and proximal regularization to have the noise-contrastive estimation (NCE) loss. The NT-Xent loss (equivalently, the InfoNCE loss (Oord et al., 2018)) is obtained by constructing a cross-entropy form loss using $2m$ augmented images from a minibatch of $m$ images (Chen et al., 2020a). Yeh et al. (2022) remove the coupling between positive and negative terms in the NT-Xent loss. Some works adaptively scale the temperature parameter (Huang et al., 2023; Manna et al., 2025; Kukleva et al., 2023). In Khosla et al. (2020), the concept of contrastive loss is applied in reverse to the supervised setting. Several studies analyze contrastive losses by decomposing them into an attracting term and a repelling term. Wang & Isola (2020) show that contrastive losses asymptotically promote alignment and uniformity in representations. Manna et al. (2021) improve performance by removing the positive–positive repulsion term and replacing the negative term with its exponential upper bound. Our work aims to help understand contrastive losses by showing how they can be derived as approximations of supervised learning objectives.

**Perspectives on SSL** There have been attempts to interpret contrastive learning within different conceptual frameworks. There is an approach that provides unified views bridging contrastive learning and covariance-based learning (Huang et al., 2021; Garrido et al., 2022; Lee et al., 2021; Balestriero & LeCun, 2022; Tian et al., 2020; Zhang et al., 2024). There is another approach that interprets contrastive learning as maximizing the mutual information of positive pairs (Hjelm et al., 2018; Oord et al., 2018; Bachman et al., 2019; Wang & Isola, 2020; Li et al., 2021; Aitchison & Ganev, 2024). HaoChen et al. (2021) views self-supervised learning as learning spectral embeddings of an augmentation graph. Beyond these analytical views, some works frame self-supervised learning from more functional viewpoints, such as clustering (Caron et al., 2020), bootstrapping (Grill et al., 2020), semi-supervised learning (Chen et al., 2020b), or knowledge distillation (Caron et al., 2021; Oquab et al., 2023). The idea of supervision is often alluded to in various approaches. We explore how self-supervised learning can be more explicitly connected to supervised learning through a principled formulation.

# 3 Problem formulation

In this section, we first formulate a supervised representation learning problem as an optimization problem, followed by its self-supervised counterpart. Throughout the paper, we use uppercase letters to denote random elements, lowercase letters to denote non-random elements (including realizations of the random elements), and calligraphic letters to denote sets.

## 3.1 Supervised representation learning problem

Let $\mathcal{X} \times \mathcal{Y}$ be a dataset comprising images and their associated visual concepts (represented as labels) of interest. To exploit the dataset to the fullest, we consider a set of transformations $\mathcal{T}$ that preserve the visual concepts and leverage them to create an augmented dataset.[2] Then, we define the augmented dataset induced by $\mathcal{T}$ as

$$
\begin{aligned}
\mathcal{T}(\mathcal{X}) \times \mathcal{Y} \\
:= \{(t(x), y) : (x, y) \in \mathcal{X} \times \mathcal{Y} \text{ and } t \in \mathcal{T}\}.
\end{aligned} \tag{1}
$$

Equipped with the augmented dataset, we want to train an encoder $f_\theta : \mathcal{X} \to \mathbb{R}^d \setminus \{0\}$ which is parameterized by learnable parameters $\theta$. It maps an image $t(x)$ to its representation $f_\theta(t(x))$. Typically, the representation dimension $d$ is small relative to the image size. By training the encoder, our goal is to make representations of images with the same visual concept, gathered close together, while representations of images with different visual concepts are meaningfully distant from each other. To keep the theoretical framework intuitive and concise, we begin with just these two fundamental ideas: positive samples are clustered, while negative samples are separated.

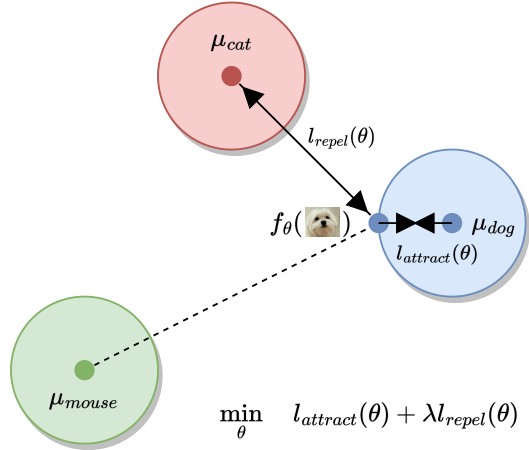

$$
\min_\theta \quad l_{attract}(\theta) + \lambda l_{repel}(\theta)
$$

Figure 1: **Supervised learning as an optimization.** The loss $l_{\text{attract}}(\theta)$ encourages the image representation to attract the prototype representation $\mu_{\text{dog}}$ that shares the visual concept of that image. On the other hand, the loss $l_{\text{repel}}(\theta)$ prompts the image representation to repel the prototype representation $\mu_{\text{cat}}$ that is closest among those not sharing the visual concept of that image. The parameter $\lambda$ balances the two losses.

To achieve our goal, we employ the concept of *prototype representation* of a visual concept to set targets for images (Li et al., 2020; Caron et al., 2020). This denotes a point in the representation space that embodies the visual concept. To see the whole approximation process, we start by assuming that an oracle gives the ideal prototype representation, which can serve as a common target for images with the same visual concept during training. However, since such an oracle does not exist in reality, we later construct the prototype representation using available data.

From now on, we tag a data point $(t(x), y) \in \mathcal{T}(\mathcal{X}) \times \mathcal{Y}$ and base the formulation on it. Let $l_{\text{attract}}(\theta)$ and $l_{\text{repel}}(\theta)$ denote the attracting and repelling components of the loss function for the image representation $f_\theta(t(x))$. Specifically, $l_{\text{attract}}(\theta)$ encourages similarity with the prototype representation $\mu_y$ of its own label, while $l_{\text{repel}}(\theta)$ penalizes similarity with the prototype representations $\mu_{y'}$ of other labels ($y' \neq y$). The similarity measure is usually chosen to be cosine similarity. Then, we formulate the supervised representation learning problem as the following optimization problem:

$$
\min_\theta \quad l_{\text{attract}}(\theta) + \lambda l_{\text{repel}}(\theta) \tag{2}
$$

where $\lambda > 0$ is a parameter which balances the two losses.

---

[2]Note that the choice of data augmentation can also be seen as a type of supervision (Xiao et al., 2020). By treating the labels of augmented images as identical, we supervise the resolution at which the model should be transformation invariant. Therefore, unlike $\mathcal{X}$, $\mathcal{T}(\mathcal{X})$ contains partial information about the labels, which enables self-supervised learning.

In contrastive learning, there is no need to repel negative samples that are already dissimilar enough. In this context, we only repel the prototype representation with the maximum similarity among those representing distinct labels. Then, our problem becomes as follows:

$$\min_{\theta} \quad -s\left(f_\theta(t(x)), \mu_y\right) + \lambda \max_{y' \neq y} s\left(f_\theta(t(x)), \mu_{y'}\right) \tag{3}$$

where $s(\cdot, \cdot)$ is a similarity measure (e.g., cosine similarity). For a better understanding, refer to Figure 1.

Note that our formulation is similar to minimizing the triplet loss in spirit (Chechik et al., 2010; Schroff et al., 2015; Schultz & Joachims, 2003; Arora et al., 2019). In our formulation, we can see $f_\theta(t(x))$ as the anchor, the prototype representation $\mu_y$ as the positive sample, and the prototype representation $\mu_{y'}$ as the negative sample. Only considering the negative sample with maximum similarity is related to the concept of hard negative mining (Girshick, 2015; Faghri et al., 2017; Oh Song et al., 2016). This idea has sometimes been implemented through the introduction of the concept of support vectors or margin (Cortes & Vapnik, 1995; Schroff et al., 2015). Pursuing this to the extreme leads us to repel the most challenging example, namely, the negative sample with maximum similarity.

Now, we construct the prototype representations. For a given label $y$, a natural choice for the prototype representation of the label is the expectation of the representations of the images with the same label, i.e.,

$$\hat{\mu}_y := \mathbb{E}_{T,X|y} f_\theta(T(X)) \tag{4}$$

where $T$ is distributed over $\mathcal{T}$, and $X$ is conditionally distributed over $\{x : (x, y) \in \mathcal{X} \times \mathcal{Y}\}$. Plugging it to Equation (3), our problem becomes as follows:

$$\min_{\theta} \quad -s\left(f_\theta(t(x)), \mathbb{E}_{T,X|y} f_\theta(T(X))\right) + \lambda \max_{y' \neq y} s\left(f_\theta(t(x)), \mathbb{E}_{T',X'|y'} f_\theta(T'(X'))\right) \tag{5}$$

where $T'$ and $X'$ are independent copies of $T$ and $X$, respectively.

## 3.2 Self-supervised representation learning problem

In the self-supervised learning regime, we do not have access to the labels. So, we use a surrogate prototype representation for the image $t(x)$ as the target. We construct it as the expectation of the representations of augmented views of the image $x$, i.e.,

$$\tilde{\mu} := \mathbb{E}_T f_\theta(T(x)). \tag{6}$$

Since data augmentation preserves labels, augmented views share the same (unobserved) label $y$. In Section 5, we demonstrate the importance of finding a data augmentation strategy that approximates well from the prototype representation $\mathbb{E}_{T,X|y} f_\theta(T(X))$ to the surrogate prototype representation $\mathbb{E}_T f_\theta(T(x))$. Plugging it in the attracting component of Equation (5), we rewrite our problem as follows:

$$\min_{\theta} \quad -s\left(f_\theta(t(x)), \tilde{\mu}\right) + \lambda \max_{y' \neq y} s\left(f_\theta(t(x)), \hat{\mu}_{y'}\right). \tag{7}$$

Note that we leave the repelling component as is since it can be managed without modification. In Section 4, we find an upper bound of the above objective function, and in Section 5, we show the upper bound can be minimized using a Siamese network. Through this, we show how attracting and repelling pseudo-labels ($\tilde{\mu}$ and $\hat{\mu}_{y'}$) can be achieved through attracting and repelling samples ($f_\theta(t'(x))$ and $f_\theta(t'(x'))$). Refer to Figure 2 for a better understanding.

# 4 Theoretical derivation

In this section, we determine upper bounds of the attracting and repelling components. Our objective is to minimize these upper bounds, addressing the optimization problem discussed in the previous section. We show that the *triplet loss with pseudo-labels* can be interpreted as an approximation to an *InfoNCE-type loss with samples*. This perspective provides a theoretical link between prototype-based supervised learning and contrastive self-supervised learning frameworks.

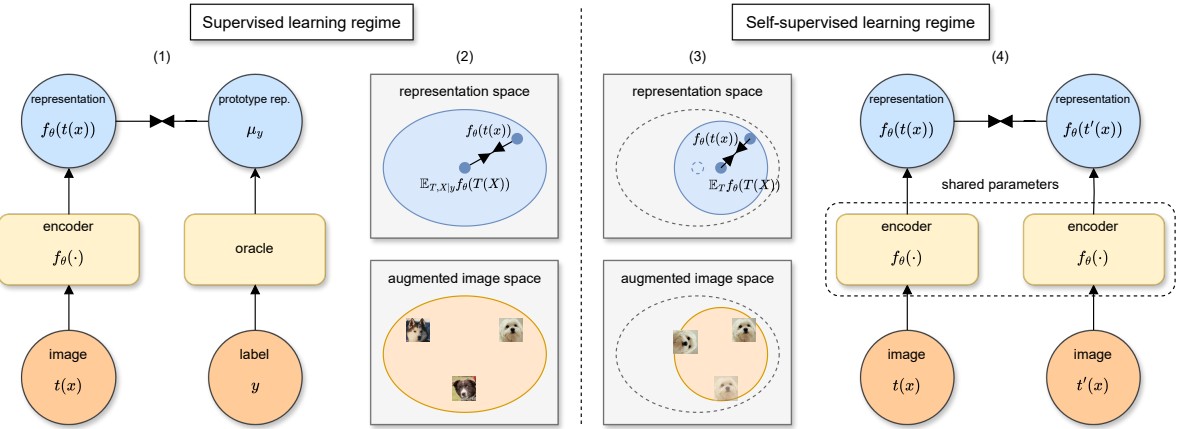

Figure 2: **Self-supervised learning as an approximation of supervised learning.** (1) In an ideal supervised regime, the ideal prototype representation $\mu_y$ is given by an oracle. (2) In a realistic supervised regime, the prototype representation is constructed as the expectation $\mathbb{E}_{T,X|y} f_\theta(T(X))$ of the representations of the images with the same label $y$. (3) In a self-supervised regime, a surrogate prototype representation is constructed as the expectation $\mathbb{E}_T f_\theta(T(x))$ of the representations of the available images sharing the same label as $t(x)$. (4) This can be effectively implemented using a Siamese network.

## 4.1 Attracting component

We first find an upper bound for the attracting component by making the following assumptions based on common practice.

**Assumption 4.1** (cosine similarity)**.** The similarity measure $s(\cdot, \cdot)$ is cosine similarity, i.e., $s(x_1, x_2) = x_1 \cdot x_2 / (\|x_1\| \|x_2\|)$. When we say $s(x_1, x_2)$, we assume $x_1$ and $x_2$ are nonzero.

**Assumption 4.2** ($l_2$-normalization)**.** Representations at the end of the encoder are $l_2$-normalized so that $\|f_\theta(t(x))\| = 1$, i.e., $f_\theta : \mathcal{X} \to \mathbb{S}^{d-1}$. Here, $\mathbb{S}^{d-1} := \{x \in \mathbb{R}^d : \|x\| = 1\}$ denotes the unit sphere in $\mathbb{R}^d$.

**Assumption 4.3** (technical assumption)**.** We additionally make a technical assumption which means that the two vectors $f_\theta(t(x))$ and $\mathbb{E}_T f_\theta(T(x))$ lie in the same hemisphere, i.e., $f_\theta(t(x)) \cdot \mathbb{E}_T f_\theta(T(x)) \geq 0$. Informally speaking, this means that the augmentation does not distort the image too much, so $\mathbb{E}_T f_\theta(T(x))$ does not point in a completely different direction.

**Theorem 4.4** (upper bound of the attracting component)**.** *Assume Assumption 4.1, 4.2, and 4.3 hold. Then,*

$$- s\left(f_\theta(t(x)), \mathbb{E}_T f_\theta(T(x))\right) \leq -\mathbb{E}_T s\left(f_\theta(t(x)), f_\theta(T(x))\right). \tag{8}$$

*Proof.* Refer to Appendix A.1.1. $\qquad\square$

We approximate the upper bound and obtain the following sample analog:

$$\widetilde{l}_{\mathrm{attract}}(\theta) := -\frac{1}{|\hat{\mathcal{T}}|} \sum_{t' \in \hat{\mathcal{T}}} s\left(f_\theta(t(x)), f_\theta(t'(x))\right) \tag{9}$$

where $\hat{\mathcal{T}}$ is the set of transformation samples.

## 4.2 Repelling component

We now find an upper bound for the repelling component by making the following assumption.

**Assumption 4.5** (balanced dataset)**.** Labels are uniformly distributed, i.e., $p(y) = \frac{1}{n}$, where $n$ is the finite number of labels.

**Theorem 4.6** (upper bound of the repelling component)**.** *Assume Assumption 4.1, 4.2, and 4.5 hold. Let $\nu := \min_{y' \neq y} \|\mathbb{E}_{T', X'|y'} f_\theta(T'(X'))\|$. Then, for all $\alpha > 0$,*

$$\max_{y' \neq y} s\left(f_\theta(t(x)), \mathbb{E}_{T', X'|y'} f_\theta(T'(X'))\right) \leq \mathbb{E}_{T'}\left[\frac{1}{\nu\alpha} \log \mathbb{E}_{X'} \exp\left(\alpha s\left(f_\theta(t(x)), f_\theta(T'(X'))\right)\right)\right] + \frac{1}{\nu\alpha} \log n. \quad (10)$$

*Proof.* We approximate the maximum function by the log-sum-exp function and apply Jensen inequality to pull out the expectations. For the detailed proof, refer to Appendix A.1.2. $\square$

If we approximate the upper bound and trim the constant terms, which are not relevant to optimization, we obtain the following:

$$\widetilde{l}_{\text{repel}}(\theta) := \frac{1}{|\hat{\mathcal{T}}|} \sum_{t' \in \hat{\mathcal{T}}} \frac{1}{\nu\alpha} \log \sum_{x' \in \hat{\mathcal{X}}} \exp(\alpha s(f_\theta(t(x)), f_\theta(t'(x')))) \quad (11)$$

where $\hat{\mathcal{T}}$ is the set of transformation samples, and $\hat{\mathcal{X}}$ is the set of image samples.

### 4.3 Total loss

By combining Equation (9) and (11), the total loss $\widetilde{l}(\theta) := \widetilde{l}_{\text{attract}}(\theta) + \lambda \widetilde{l}_{\text{repel}}(\theta)$ is as follows:

$$\widetilde{l}(\theta) = \frac{1}{|\hat{\mathcal{T}}|} \sum_{t' \in \hat{\mathcal{T}}} \left[-s\left(f_\theta(t(x)), f_\theta(t'(x))\right) + \frac{\lambda}{\nu}\left[\frac{1}{\alpha} \log \sum_{x' \in \hat{\mathcal{X}}} \exp(\alpha s(f_\theta(t(x)), f_\theta(t'(x'))))\right]\right]. \quad (12)$$

By rearranging, we have

$$\widetilde{l}(\theta) = \frac{1}{\alpha|\hat{\mathcal{T}}|} \sum_{t' \in \hat{\mathcal{T}}} \left[-\log \frac{\exp(\alpha s\left(f_\theta(t(x)), f_\theta(t'(x))\right))}{\left(\sum_{x' \in \hat{\mathcal{X}}} \exp(\alpha s(f_\theta(t(x)), f_\theta(t'(x'))))\right)^{\lambda/\nu}}\right]. \quad (13)$$

Note that this equation and the NT-Xent in SimCLR are similar in their forms, which we discuss in more detail in the next section.

## 5 Theoretical insights

In this section, we present theoretical insights derived from our framework, illustrating how it relates to several components commonly used in self-supervised learning. We use SimCLR (Chen et al., 2020a) as a primary example, as it has served as a central reference point for many subsequent algorithms.

For our experiments, we adopt SimCLR with a temperature parameter $\tau = 0.5$, using ImageNet (Deng et al., 2009) as the dataset and ResNet-50 (He et al., 2016) as the backbone. We assess top-1 accuracy using linear evaluation, a standard protocol for evaluating self-supervised learning algorithms. For a fair comparison, all settings are kept the same except for the specific factor under investigation. For the detailed implementation, refer to A.3.

### 5.1 Loss: NT-Xent

Let $\{x_1, \ldots, x_m\}$ be a minibatch of $m$ images. If we transform each image in two different ways and pass them through the encoder, we obtain representation pairs $\{(f_\theta(t(x_i)), f_\theta(t'(x_i))) : i = 1, \ldots, m\}$ of $2m$ augmented

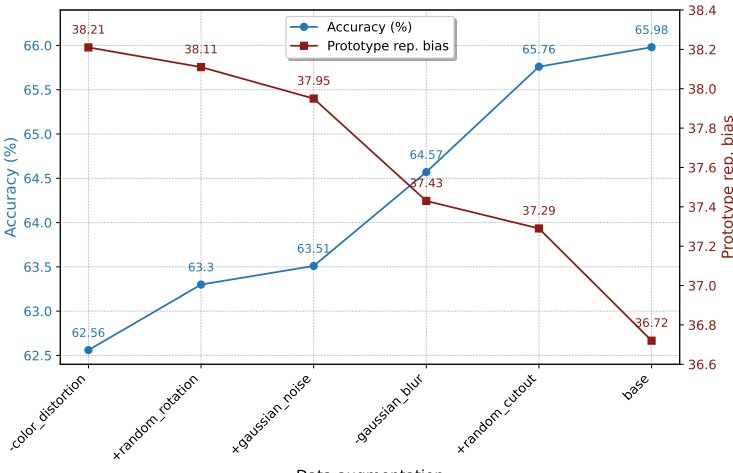

Figure 3: **Accuracy vs. prototype representation bias.** We investigate the relationship between accuracy and prototype representation bias by adding or removing transformations from SimCLR's data augmentation strategy (base). Lower prototype representation bias tends to result in higher accuracy.

images, which we denote as $\{(z_i, z_i') : i = 1, \ldots, m\}$. Then, in the case of $\lambda = \nu$, the summand in Equation (13) can be implemented as

$$-\log \frac{\exp(\alpha s(z_i, z_i'))}{\sum_{j \in [m] \setminus \{i\}} \exp(\alpha s(z_i, z_j'))} \tag{14}$$

where $[m] := \{1, \ldots, m\}$.

On the other hand, in the NT-Xent loss used in SimCLR, if we let the temperature parameter $\tau$ be $1/\alpha$, the NT-Xent loss is represented as

$$-\log \frac{\exp(\alpha s(z_i, z_i'))}{\sum_{j \in [m]} \exp(\alpha s(z_i, z_j')) + \sum_{j \in [m] \setminus \{i\}} \exp(\alpha s(z_i, z_j))}. \tag{15}$$

This is a variant of Equation (14). Having the second summation in the denominator can be seen as a method to more fully exploit the provided representations, since $(z_i, z_j)$ are also considered negative pairs when $j \neq i$.

In the first summation in the denominator, the positive pair is explicitly excluded in our theoretical derivation, yielding a decoupled loss formulation. Interestingly, this coincides with the decoupled contrastive loss proposed by Yeh et al. (2022), who empirically showed that summing over $[m] \setminus \{i\}$ performs better than over $[m]$.

Common expressions of contrastive losses, such as cross-entropy and temperature, typically frame them in the form of the Boltzmann (or Gibbs) distribution. Our framework offers a complementary perspective by deriving a similar structure from a supervised learning formulation.

## 5.2 Data augmentation: debiased prototype representation

When transitioning from supervised to self-supervised learning, we approximate the prototype representation $\mathbb{E}_{T,X|y} f_\theta(T(X))$ with the surrogate prototype representation $\mathbb{E}_T f_\theta(T(x))$. To examine the quality of this approximation, we define the *prototype representation bias* as

$$\text{Bias}_{\text{proto}} := \mathbb{E}_{(X_0, Y_0)} \|\mathbb{E}_{T, X|Y_0} f_\theta(T(X)) - \mathbb{E}_T f_\theta(T(X_0))\|. \tag{16}$$

We hypothesize that reducing this bias is associated with improved downstream accuracy. To test this, we vary the distribution of $T$ through different data augmentation strategies. Specifically, we compare SimCLR's

default data augmentation (`base`) with cases where we exclude Gaussian blur (`-gaussian_blur`) and color distortion (`-color_distortion`), and with cases where we include random cutout (`+random_cutout`), random rotation (`+random_rotation`), and gaussian noise (`+gaussian_noise`), resulting in a total of six scenarios.

Figure 3 shows that using data augmentation with debiased prototype representation leads to an increase in accuracy. Notably, SimCLR's default augmentation achieves both the highest accuracy and the smallest bias. Interestingly, enriching the data augmentation by adding transformations such as random cutout, random rotation, or gaussian noise does not improve accuracy. This may be due to an increased mismatch between the surrogate and true prototype representations.

### 5.3 Similarity measure: cosine similarity with normalized representations

When computing similarity between two representations, many self-supervised learning algorithms including SimCLR normalize the representations and calculate cosine similarity as in Assumption 4.1 and 4.2. To investigate the empirical implications of these assumptions, we compare three cases: 1) cosine similarity with normalization, 2) dot product without normalization, and 3) negative Euclidean distance without normalization.[3]

Table 1: Comparison of similarity measures with and without $l_2$-normalization. The results show that cosine similarity with normalization significantly outperforms the other variants.

| CS w/ $l_2$ | Dot w/o $l_2$ | -Eucl. w/o $l_2$ |
|---|---|---|
| 65.98 | 0.43 | 10.63 |

Table 1 shows that cosine similarity with normalized representations significantly outperforms the alternatives. Among the unnormalized variants, negative Euclidean distance performs better than the dot product, possibly because it captures spatial dissimilarity more directly. These results suggest that the widespread use of cosine similarity with normalization in contrastive learning is consistent with both empirical effectiveness and the assumptions required for tractable theoretical analysis.

### 5.4 Dataset: balanced class distribution

To examine the effect of class balance as in Assumption 4.5, we conduct a controlled experiment comparing uniform and long-tailed class distributions. In both cases, the training sets contain the same number of images (115,846, which is 9% of the ImageNet training set), but they differ in class distribution. We use an identical test set for both cases.

Table 2: Comparison of class distributions. The results show that the uniform class distribution leads to better performance.

| Uniform | Long-tailed |
|---|---|
| 20.82 | 13.65 |

Table 2 shows that SimCLR performs better on a balanced dataset compared to an imbalanced one. The observed effect supports the idea that class balance, a widely adopted practice in contrastive learning (Assran et al., 2022b;a; Zhou et al., 2022), aligns with assumptions that enable tractable theoretical analysis in our framework.

### 5.5 Architecture: Siamese networks

The upper bound $-\mathbb{E}_T s\left(f_\theta(t(x)), f_\theta(T(x))\right)$ in Equation (8) involves comparing the similarity between two representations $f_\theta(t(x))$ and $f_\theta(t'(x))$, where $t$ and $t'$ are independently sampled augmentations. This naturally corresponds to a Siamese network architecture (Bromley et al., 1993), where a single image $x$ is augmented twice to produce $t(x)$ and $t'(x)$, and each is passed through a shared encoder $f_\theta$. Siamese networks naturally align with the structure of similarity-based objectives in our framework.

Although Siamese networks are typically symmetric, with two encoders that share parameters and have identical architectures, several algorithms introduce asymmetry to improve performance (He et al., 2020;

---

[3]Note that when dealing with two normalized vectors, cosine similarity is equivalent to the dot product. Additionally, negative Euclidean distance with normalization is equivalent to cosine similarity with normalization since $-\|a - b\|^2 = -2 + 2a \cdot b$.

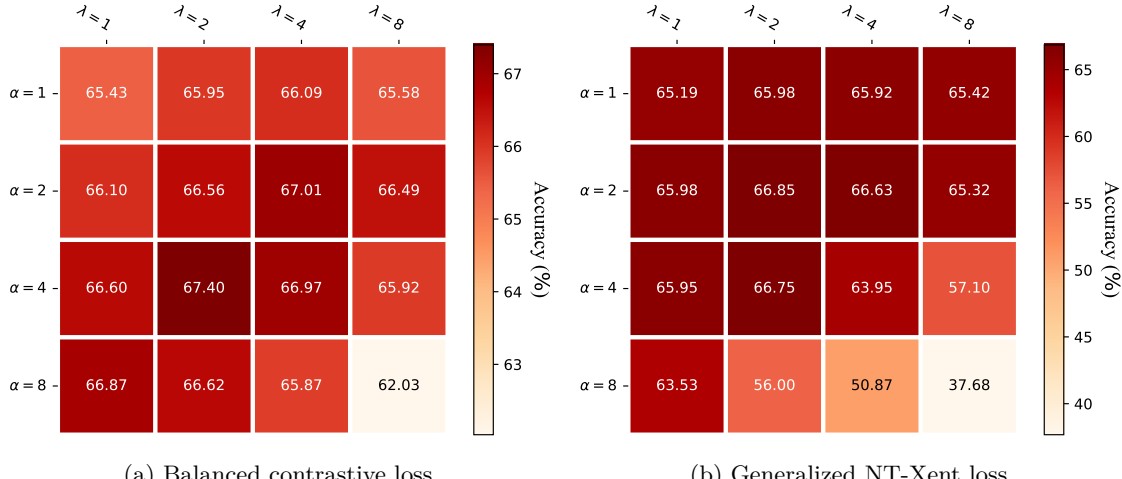

(a) Balanced contrastive loss            (b) Generalized NT-Xent loss

Figure 4: **Impact of balancing parameters $\alpha$ and $\lambda$.** Better balancing can be accomplished through the adjustments of the balancing parameters.

Chen & He, 2021; Grill et al., 2020; Caron et al., 2020; 2021; Oquab et al., 2023; Tian et al., 2021). In such cases, it has been empirically observed that performance improves when one encoder produces outputs with lower variance than the other (Wang et al., 2022). The lower-variance encoder is commonly referred to as the target or teacher, and the higher-variance encoder as the source or student.

In our problem formulation, the original attracting component in Equation (8) is $-s\left(f_\theta(t(x)), \mathbb{E}_T f_\theta(T(x))\right)$ where the two attracting objects $f_\theta(t(x))$ and $\mathbb{E}_T f_\theta(T(x))$ are asymmetric. Note that $\mathbb{E}_T f_\theta(T(x))$ can be approximated by $\frac{1}{n}\sum_{i=1}^n f_\theta(T_i(x))$, and $\frac{1}{n}\sum_{i=1}^n f_\theta(T_i(x))$ has less variance than $f_\theta(T(x))$.

This suggests that our problem formulation, along with Theorem 4.4, may provide insight into the coexistence of both symmetric and asymmetric designs in the self-supervised learning literature.

## 6 Empirical study

In this section, we introduce a loss that is motivated by the form of Equation (12). Our aim is to help understand the roles of the balancing parameters that constitute this loss in our framework and to empirically report how varying them affects performance. For notational simplicity, we rewrite $\lambda/\nu$ as $\lambda$. Given a representation $z$ among the $2m$ representations obtained from a minibatch of $m$ images, we define the following loss:

$$-s(z, z^+) + \lambda\left[\frac{1}{\alpha}\log\sum_{z^-}\exp(\alpha s(z, z^-))\right] \tag{17}$$

where $(z, z^+)$ is the positive pair and $(z, z^-)$ are $2(m-1)$ negative pairs. The cost for the whole minibatch is then calculated by taking the mean of the losses of all representations. Note that the attracting component consists of one attracting force, and the repelling component consists of multiple repelling forces. We refer to this as the *balanced contrastive loss*.

There are two hyperparameters $\alpha > 0$ and $\lambda > 0$ in the balanced contrastive loss. We refer to these as the *balancing parameters* since each governs a different form of balance in contrastive learning. The parameter $\alpha$ modulates the relative influence among negative samples within the repelling term (Kalantidis et al., 2020; Zhang et al., 2022; Jiang et al., 2024). Note that the repelling component is a smooth approximation to the maximum function (refer to Lemma A.1 and Wang & Liu (2021)):

$$\lim_{\alpha\to\infty}\left[\frac{1}{\alpha}\log\sum_{z^-}\exp(\alpha s(z, z^-))\right] = \max_{z^-} s(z, z^-). \tag{18}$$

As $\alpha$ increases, representations with higher similarity contribute more strongly to the repelling term. In self-supervised learning, negative samples may include images with the same label (referred to as sampling bias in Chuang et al. (2020)). So, if $\alpha$ is too large, there is a risk of repelling images with the same label. Appropriately choosing $\alpha$ can be interpreted as a form of risk hedging over multiple negative samples. This also offers insight into the role of the temperature parameters of InfoNCE-type losses. On the other hand, the parameter $\lambda$ adjusts the relative magnitudes of the attracting and repelling forces.

To investigate the impact of balancing parameters $\alpha$ and $\lambda$, we evaluate the balanced contrastive loss over a grid of parameters $\{(\alpha, \lambda) : \alpha, \lambda \in \{1, 2, 4, 8\}\}$. We also consider a variant where the positive pair is included in the repelling component in Equation (17), which we refer to as the *generalized NT-Xent loss*, as it reduces to NT-Xent when $\lambda = 1$. Figure 4 illustrates the changes in accuracy based on various combinations of the parameters. Note that, since ImageNet contains 1,000 classes, the chance-level top-1 accuracy is 0.1%.

Overall, the balanced contrastive loss achieves higher peak performance than the generalized NT-Xent loss. For the balanced contrastive loss, the best performance is obtained at $(\alpha, \lambda) = (4, 2)$, while the generalized NT-Xent loss performs best at $(2, 2)$. In both cases, the highest accuracy is not achieved when $\lambda = 1$. This highlights the significance of the balancing parameter $\lambda$. Additionally in both scenarios, it is crucial for $\alpha$ to have an appropriate value that is not too large or too small. Specifically for the generalized NT-Xent, it is advantageous to set $\alpha$ to a smaller value compared to the balanced contrastive loss. This may be due to the presence of the positive sample in the repelling component, meaning that increasing $\alpha$ results in a larger repulsion of the positive sample.

Given the 0.1% chance-level accuracy on ImageNet, these performance differences are substantial, especially considering they are achieved solely by adjusting the balancing parameters. These results suggest that further improvements to contrastive losses may be possible through better balancing.

## 7 Conclusion

In this work, we present a theoretical framework that conceptualizes self-supervised representation learning as an approximation to supervised representation learning. Starting from a concise formulation of the supervised objective, we derive how a natural approximation emerges in the absence of labels. In particular, we show that the triplet loss with pseudo-labels can be viewed as an approximation to an InfoNCE-type loss with samples, offering a principled explanation for the structure of widely used contrastive losses. Our framework provides theoretical insights into common design choices in self-supervised learning. Additionally, it sheds light on sources of bias in prototype representations and motivates a balanced contrastive loss that improves empirical performance. We hope that our work will benefit the community by offering helpful perspectives and encouraging further exploration of the connections between supervised and self-supervised learning.

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

# A   Appendix

## A.1   Proofs

This subsection presents the proofs of Theorem 4.4 and Theorem 4.6.

### A.1.1   Proof of Theorem 4.4

We restate the assumptions and the theorem and provide the proof below.

**Assumption 4.1** (cosine similarity). The similarity measure $s(\cdot,\cdot)$ is cosine similarity, i.e., $s(x_1, x_2) = x_1 \cdot x_2/(\|x_1\|\|x_2\|)$. When we say $s(x_1, x_2)$, we assume $x_1$ and $x_2$ are nonzero.

**Assumption 4.2** ($l_2$-normalization). Representations at the end of the encoder are $l_2$-normalized so that $\|f_\theta(t(x))\| = 1$, i.e., $f_\theta : \mathcal{X} \to \mathbb{S}^{d-1}$. Here, $\mathbb{S}^{d-1} := \{x \in \mathbb{R}^d : \|x\| = 1\}$ denotes the unit sphere in $\mathbb{R}^d$.

**Assumption 4.3** (technical assumption). We additionally make a technical assumption which means that the two vectors $f_\theta(t(x))$ and $\mathbb{E}_T f_\theta(T(x))$ lie in the same hemisphere, i.e., $f_\theta(t(x)) \cdot \mathbb{E}_T f_\theta(T(x)) \geq 0$. Informally speaking, this means that the augmentation does not distort the image too much, so $\mathbb{E}_T f_\theta(T(x))$ does not point in a completely different direction.

**Theorem 4.4** (upper bound of the attracting component). *Assume Assumption 4.1, 4.2, and 4.3 hold. Then,*

$$- s\left(f_\theta(t(x)), \mathbb{E}_T f_\theta(T(x))\right) \leq -\mathbb{E}_T s\left(f_\theta(t(x)), f_\theta(T(x))\right). \tag{8}$$

*Proof.*

$$-s\left(f_\theta(t(x)), \mathbb{E}_T f_\theta(T(x))\right) \overset{(i)}{=} -\frac{f_\theta(t(x)) \cdot \mathbb{E}_T f_\theta(T(x))}{\|f_\theta(t(x))\|\|\mathbb{E}_T f_\theta(T(x))\|} \tag{19}$$

$$\overset{(ii)}{=} -\frac{f_\theta(t(x)) \cdot \mathbb{E}_T f_\theta(T(x))}{\|\mathbb{E}_T f_\theta(T(x))\|} \tag{20}$$

$$\overset{(iii)}{\leq} -\frac{f_\theta(t(x)) \cdot \mathbb{E}_T f_\theta(T(x))}{\mathbb{E}_T \|f_\theta(T(x))\|} \tag{21}$$

$$\overset{(iv)}{=} -f_\theta(t(x)) \cdot \mathbb{E}_T f_\theta(T(x)) \tag{22}$$

$$\overset{(v)}{=} -\mathbb{E}_T \left[f_\theta(t(x)) \cdot f_\theta(T(x))\right] \tag{23}$$

$$\overset{(vi)}{=} -\mathbb{E}_T \left[\frac{f_\theta(t(x)) \cdot f_\theta(T(x))}{\|f_\theta(t(x))\|\|f_\theta(T(x))\|}\right] \tag{24}$$

$$\overset{(vii)}{=} -\mathbb{E}_T s\left(f_\theta(t(x)), f_\theta(T(x))\right) \tag{25}$$

where $(i)$ and $(vii)$ are by Assumption 4.1, $(ii)$, $(iv)$, and $(vi)$ are by Assumption 4.2, $(iii)$ is by Assumption 4.3, the convexity of $l^2$-norm (Boyd & Vandenberghe, 2004), and Jensen's inequality, and $(v)$ is by the linearity of expectation. This completes the proof of Theorem 4.4. □

### A.1.2   Proof of Theorem 4.6

Before we prove Theorem 4.6, we need three additional lemmas. While the proofs of the lemmas are straightforward, they are not readily available in the existing literature. Therefore, we provide them here for the sake of self-containedness.

**Lemma A.1.** *For $\alpha > 0$ and $x_i \in \mathbb{R}$, $i = 1, 2, \ldots, n$,*

$$\max_{i=1,\ldots,n} x_i \leq (1/\alpha) \log \sum_{i=1}^{n} \exp(\alpha x_i) \leq \max_{i=1,\ldots,n} x_i + \frac{\log n}{\alpha}, \tag{26}$$

*where the equalities hold when $\alpha$ goes to infinity.*

*Proof.* We have

$$\exp\left(\max_{i=1,\ldots,n}(\alpha x_i)\right) \le \sum_{i=1}^{n}\exp(\alpha x_i) \le n\exp\left(\max_{i=1,\ldots,n}(\alpha x_i)\right). \tag{27}$$

Since $\alpha > 0$,

$$\alpha\max_{i=1,\ldots,n}x_i \le \log\sum_{i=1}^{n}\exp(\alpha x_i) \le \alpha\max_{i=1,\ldots,n}x_i + \log n. \tag{28}$$

This completes the proof of Lemma A.1. $\square$

**Lemma A.2.** *For $\alpha > 0$ and $x_i \in \mathbb{R}$, $i = 1, 2, \ldots, n$,*

$$u(x_1,\ldots,x_n) := (1/\alpha)\log\sum_{i=1}^{n}\exp(\alpha x_i) \tag{29}$$

*is convex on $\mathbb{R}^n$.*

*Proof.* Note that the log-sum-exp function $v(x_1,\ldots,x_n) := \log\sum_{i=1}^{n}\exp(x_i)$ is convex on $\mathbb{R}^n$ (Boyd & Vandenberghe, 2004; Ghaoui, 2014). $u(x_1,\ldots,x_n) = (1/\alpha)v(\alpha(x_1,\ldots,x_n))$, and composition with an affine mapping preserves convexity (Boyd & Vandenberghe, 2004). Thus, $u(x_1,\ldots,x_n)$ is also convex on $\mathbb{R}^n$. This completes the proof of Lemma A.2. $\square$

**Lemma A.3.** *If $g_1(x) \ge 0$ for all $x$, and $g_2(x) \ge 0$ for some $x$, then*

$$\max[g_1(x)g_2(x)] \le \max[g_1(x)]\max[g_2(x)]. \tag{30}$$

*Proof.* By default, $g_2(x) \le \max[g_2(x)]$. Since $g_1(x) \ge 0$ for all $x$, $g_1(x)g_2(x) \le g_1(x)\max[g_2(x)]$. Taking the maximum of both sides, we have $\max[g_1(x)g_2(x)] \le \max[g_1(x)\max[g_2(x)]]$. Since $g_2(x) \ge 0$ for some $x$, $\max[g_2(x)] \ge 0$, and thus $\max[g_1(x)g_2(x)] \le \max[g_1(x)]\max[g_2(x)]$. This completes the proof of Lemma A.3. $\square$

Now, we are ready to prove Theorem 4.6. We restate the assumption and the theorem and provide the proof below.

**Assumption 4.5** (balanced dataset)**.** Labels are uniformly distributed, i.e., $p(y) = \frac{1}{n}$, where $n$ is the finite number of labels.

**Theorem 4.6** (upper bound of the repelling component)**.** *Assume Assumption 4.1, 4.2, and 4.5 hold. Let $\nu := \min_{y'\ne y}\|\mathbb{E}_{T',X'|y'}f_\theta(T'(X'))\|$. Then, for all $\alpha > 0$,*

$$\max_{y'\ne y}s\left(f_\theta(t(x)),\mathbb{E}_{T',X'|y'}f_\theta(T'(X'))\right) \le \mathbb{E}_{T'}\left[\frac{1}{\nu\alpha}\log\mathbb{E}_{X'}\exp\left(\alpha s\left(f_\theta(t(x)),f_\theta(T'(X'))\right)\right)\right] + \frac{1}{\nu\alpha}\log n. \tag{10}$$

*Proof.*

$$\max_{y'\ne y}s\left(f_\theta(t(x)),\mathbb{E}_{T',X'|y'}f_\theta(T'(X'))\right) \overset{(i)}{=} \max_{y'\ne y}\frac{f_\theta(t(x))\cdot\mathbb{E}_{T',X'|y'}f_\theta(T'(X'))}{\|f_\theta(t(x))\|\|\mathbb{E}_{T',X'|y'}f_\theta(T'(X'))\|} \tag{31}$$

$$\overset{(ii)}{=} \max_{y'\ne y}\frac{f_\theta(t(x))\cdot\mathbb{E}_{T',X'|y'}f_\theta(T'(X'))}{\|\mathbb{E}_{T',X'|y'}f_\theta(T'(X'))\|} \tag{32}$$

$$\overset{(iii)}{\le} \frac{1}{\nu}\max_{y'\ne y}\mathbb{E}_{T',X'|y'}s\left(f_\theta(t(x)),f_\theta(T'(X'))\right) \tag{33}$$

where $(i)$ is by Assumption 4.1, $(ii)$ is by Assumption 4.2, and $(iii)$ is by the following argument.

Let $y^*$ be the label that achieves the maximum in Equation (32). Note that under Assumption 4.2, $0 < \|\mathbb{E}_{T',X'|y'}f_\theta(T'(X'))\| \le 1$. If in an ideal case, $f_\theta(t'(x'))$ produces the same representation for every $t'(x')$ that shares the same label $y'$, then $\|\mathbb{E}_{T',X'|y'}f_\theta(T'(X'))\| = \|f_\theta(t'(x'))\| = 1$. To show $(iii)$, we proceed by considering the following two cases.

Case 1: If $f_\theta(t(x)) \cdot \mathbb{E}_{T',X'|y*} f_\theta(T'(X')) \leq 0$, then

$$\frac{f_\theta(t(x)) \cdot \mathbb{E}_{T',X'|y*} f_\theta(T'(X'))}{\|\mathbb{E}_{T',X'|y*} f_\theta(T'(X'))\|} \overset{(i)}{\leq} \frac{f_\theta(t(x)) \cdot \mathbb{E}_{T',X'|y*} f_\theta(T'(X'))}{\mathbb{E}_{T',X'|y*} \|f_\theta(T'(X'))\|} \tag{34}$$

$$\overset{(ii)}{=} f_\theta(t(x)) \cdot \mathbb{E}_{T',X'|y*} f_\theta(T'(X')) \tag{35}$$

$$\overset{(iii)}{=} \mathbb{E}_{T',X'|y*} s(f_\theta(t(x)), f_\theta(T'(X'))) \tag{36}$$

$$\leq \max_{y' \neq y} \mathbb{E}_{T',X'|y'} s(f_\theta(t(x)), f_\theta(T'(X'))) \tag{37}$$

$$\overset{(iv)}{\leq} \frac{1}{\nu} \max_{y' \neq y} \mathbb{E}_{T',X'|y'} s(f_\theta(t(x)), f_\theta(T'(X'))) \tag{38}$$

where $(i)$ is by Jensen's inequality, $(ii)$ is by Assumption 4.2, $(iii)$ is by a similar argument in the proof of Theorem 4.4, and $(iv)$ follows from the fact that $0 < \nu \leq 1$.

Case 2: If $f_\theta(t(x)) \cdot \mathbb{E}_{T',X'|y*} f_\theta(T'(X')) > 0$, then

$$\frac{f_\theta(t(x)) \cdot \mathbb{E}_{T',X'|y*} f_\theta(T'(X'))}{\|\mathbb{E}_{T',X'|y*} f_\theta(T'(X'))\|} \overset{(i)}{\leq} \max_{y' \neq y} \frac{1}{\|\mathbb{E}_{T',X'|y'} f_\theta(T'(X'))\|} \max_{y' \neq y} \left[ f_\theta(t(x)) \cdot \mathbb{E}_{T',X'|y'} f_\theta(T'(X')) \right] \tag{39}$$

$$= \frac{1}{\nu} \max_{y' \neq y} \left[ f_\theta(t(x)) \cdot \mathbb{E}_{T',X'|y'} f_\theta(T'(X')) \right] \tag{40}$$

$$\overset{(ii)}{=} \frac{1}{\nu} \max_{y' \neq y} \mathbb{E}_{T',X'|y'} s(f_\theta(t(x)), f_\theta(T'(X'))) \tag{41}$$

where $(i)$ is by Lemma A.3, and $(ii)$ is by a similar argument in the proof of Theorem 4.4.

Now for brevity, let $g(T'(X')) := s\left(f_\theta(t(x)), f_\theta(T'(X'))\right)$. Then,

$$\max_{y' \neq y} \mathbb{E}_{T',X'|y'} g(T'(X')) \overset{(i)}{\leq} \frac{1}{\alpha} \log \sum_{y' \neq y} \exp\left(\alpha \mathbb{E}_{T',X'|y'} g(T'(X'))\right) \tag{42}$$

$$\overset{(ii)}{\leq} \frac{1}{\alpha} \log \sum_{y'} \exp\left(\alpha \mathbb{E}_{T',X'|y'} g(T'(X'))\right) \tag{43}$$

$$= \frac{1}{\alpha} \log \sum_{y'} \exp\left(\alpha \mathbb{E}_{T'} \mathbb{E}_{X'|y'} g(T'(X'))\right) \tag{44}$$

$$\overset{(iii)}{\leq} \mathbb{E}_{T'} \left[ \frac{1}{\alpha} \log \sum_{y'} \exp\left(\alpha \mathbb{E}_{X'|y'} g(T'(X'))\right) \right] \tag{45}$$

$$\overset{(iv)}{\leq} \mathbb{E}_{T'} \left[ \frac{1}{\alpha} \log \sum_{y'} \mathbb{E}_{X'|y'} \exp\left(\alpha g(T'(X'))\right) \right] \tag{46}$$

$$\overset{(v)}{=} \mathbb{E}_{T'} \left[ \frac{1}{\alpha} \log \left( n \sum_{y'} p(y') \mathbb{E}_{X'|y'} \exp\left(\alpha g(T'(X'))\right) \right) \right] \tag{47}$$

$$= \mathbb{E}_{T'} \left[ \frac{1}{\alpha} \log \left( n \mathbb{E}_{Y'} \mathbb{E}_{X'|Y'} \exp\left(\alpha g(T'(X'))\right) \right) \right] \tag{48}$$

$$= \mathbb{E}_{T'} \left[ \frac{1}{\alpha} \log \left( n \mathbb{E}_{X'} \exp\left(\alpha g(T'(X'))\right) \right) \right] \tag{49}$$

$$= \mathbb{E}_{T'} \left[ \frac{1}{\alpha} \log \left( \mathbb{E}_{X'} \exp\left(\alpha g(T'(X'))\right) \right) \right] + \frac{1}{\alpha} \log n. \tag{50}$$

where $(i)$ is by Lemma A.1, $(ii)$ is by the positivity of $\exp(\alpha x)$ and the monotonicity of $\log(x)$, $(iii)$ is by Lemma A.2 and Jensen's inequality, $(iv)$ is by the convexity of $\exp(\alpha x)$, Jensen's inequality, and the monotonicity of $\log(x)$, and $(v)$ is by Assumption 4.5. This completes the proof of Theorem 4.6. $\qquad \square$

### A.2 Cross-reference

Table 3 shows how each component of SimCLR corresponds to specific parts of our problem formulation and theoretical derivation.

Table 3: Cross-reference between SimCLR and our framework. We compare the key components and provide references to the corresponding sections and theorems.

| Component | SimCLR | Our framework |
|---|---|---|
| Architecture | Siamese network | Subsection 4.1 and 4.2 |
| Loss | NT-Xent | Subsection 4.3 |
| Data augmentation | debiased prototype representation | Subsection 3.2 |
| Similarity measure | cosine similarity with normalization | Theorem 4.4 and 4.6 |
| Dataset | balanced class distribution | Theorem 4.6 |

### A.3 Implementation details

This subsection offers a comprehensive description of the implementation details for our experiments. Readers can also refer to the code provided in the supplementary material. With 8 NVIDIA V100 GPUs, the pretraining takes about 2.5 days and 13 GB peak memory usage, the linear evaluation takes about 1.5 days and 8 GB peak memory usage, and the $k$-nearest neighbors takes about 40 minutes and 30 GB peak memory usage.

#### A.3.1 Base setting

**Dataset**  We use ImageNet as the benchmark dataset, as it is one of the most representative large-scale image datasets. The training set comprises 1,281,167 images, while the validation set comprises 50,000 images. As ImageNet's test set labels are unavailable, we utilize the validation set as a test set for evaluation purposes. ImageNet encompasses 1,000 classes.

**Data augmentation**  The following data transformations are sequentially applied during pretraining. Due to variations in image sizes, they are first cropped to dimensions of $224 \times 224$.

- `RandomResizedCrop`: Randomly crop a patch of the image within the scale range of $(0.2, 1)$, then resize it to dimensions of $(224, 224)$.

- `ColorJitter`: Change the image's brightness, contrast, saturation, and hue with strengths of $(0.4, 0.4, 0.4, 0.1)$ with a probability of $0.8$.

- `RandomGrayscale`: Convert the image to grayscale with a probability of $0.2$.

- `GaussianBlur`: Apply the Gaussian blur filter to the image with a radius sampled uniformly from the range $[0.1, 2]$ with a probability of $0.5$.

- `RandomHorizontalFlip`: Horizontally flip the image with a probability of $0.5$.

- `Normalize`: Normalize the image using a mean of $(0.485, 0.456, 0.406)$ and a standard deviation of $(0.229, 0.224, 0.225)$.

**Network architecture**  The encoder consists of a backbone followed by a projector. We employ ResNet-50 as the backbone and a three-layered fully-connected MLP as the projector. For the projector, the input and output dimensions of all layers are set to 2,048. Batch normalization (Ioffe & Szegedy, 2015) is applied to all layers, and the ReLU activation function is applied to the first two layers.

**Pretraining configuration**  We pretrain the encoder with a batch size of 512 for 100 epochs. We employ the SGD optimizer and set the momentum to 0.9, the learning rate to 0.1, and the weight decay rate to 0.0001. Additionally, we implement a cosine decay schedule for the learning rate, as proposed by Loshchilov & Hutter (2016); Chen et al. (2020a).

**Evaluation configuration**   After pretraining, we employ linear evaluation, which is the standard evaluation protocol. We take and freeze the pretrained backbone and attach a linear classifier on top. The linear classifier is then trained on the training set and evaluated on the test set. Training the linear classifier is conducted with a batch size of 4,096 for 90 epochs, utilizing the LARS optimizer (You et al., 2017).

### A.3.2   Implementation details for Section 5.2

To estimate the value of the prototype representation bias, for each $(x_i, y_i)$ in the ImageNet training set $\mathcal{D}$, we sample $t_i$ from $T$ and $x_i'$ from $X|y_i$ and calculate the deviation $\|f_\theta(t_i(x_i')) - f_\theta(t_i(x_i))\|$. Then, we take the average over the entire $\mathcal{D}$ as follows:

$$\frac{1}{|\mathcal{D}|} \sum_{(x_i, y_i) \in \mathcal{D}} \|f_\theta(t_i(x_i')) - f_\theta(t_i(x_i))\|. \tag{51}$$

So, we consider total 1,281,167 samples, which is equivalent to the number of images in the ImageNet training set.

### A.3.3   Implementation details for Section 5.3

When normalization is not carried out, there is a risk of loss overflow, so we resort to using the log-sum-exp trick. It does not alter the values themselves.

### A.3.4   Implementation details for Section 5.4

We use ImageNet-LT (ImageNet Long-Tailed) as a benchmark for imbalanced datasets. ImageNet-LT is a representative dataset specifically designed to address the challenges associated with imbalanced datasets. It is subsampled across the 1,000 classes of ImageNet, following a Pareto distribution with a shape parameter $\alpha$ of 6. The training set consists of 115,846 images, which is approximately 9% of the entire ImageNet training set. The class with the most images contains 1,280 images, while the class with the fewest has only 5 images. The test set is balanced, consisting of 50,000 images, with each class having exactly 50 images.

We construct ImageNet-Uni (ImageNet Uniform) as a subset of ImageNet to enable a fair comparison. We uniformly sample 115,846 images from the ImageNet training set, matching the size of the ImageNet-LT training set. The test set is configured to be identical to that of ImageNet-LT.

### A.4   Further experiments

In this subsection, we provide additional experimental results. We include results on CIFAR-10 (Krizhevsky et al., 2009). Note that, since CIFAR-10 contains 10 classes, the chance-level accuracy is 10%.

### A.4.1   Implementation details for CIFAR-10 experiments

**Dataset**   The training set comprises 50,000 images, while the test set comprises 10,000 images. CIFAR-10 contains 10 classes, with all images standardized to a fixed size of $32 \times 32$.

**Data augmentation**   The following data transformations are sequentially applied during pretraining.

- `RandomResizedCrop`: Randomly crop a patch of the image within the scale range of $(0.08, 1)$, then resize it to dimensions of $(32, 32)$.

- `RandomHorizontalFlip`: Horizontally flip the image with a probability of 0.5.

- `ColorJitter`: Change the image's brightness, contrast, saturation, and hue with strengths of $(0.4, 0.4, 0.4, 0.1)$ with a probability of 0.8.

- `RandomGrayscale`: Convert the image to grayscale with a probability of 0.2.

- `Normalize`: Normalize the image using a mean of $(0.485, 0.456, 0.406)$ and a standard deviation of $(0.229, 0.224, 0.225)$.

Table 4: Standard evaluations. We report top-1 accuracy on CIFAR-10 and ImageNet using two standard evaluation protocols: $k$-nearest neighbor and linear evaluation. Each result is presented as the mean $\pm$ standard deviation over 5 runs.

| Dataset | Protocol | |
|---|---|---|
| | $k$-NN | Linear eval. |
| CIFAR-10 | $80.32 \pm 0.32$ | $86.08 \pm 0.07$ |
| ImageNet | $51.00 \pm 0.22$ | $67.40 \pm 0.07$ |

(a) Balanced contrastive loss

(b) Generalized NT-Xent loss

Figure 5: **Impact of balancing parameters $\alpha$ and $\lambda$ on CIFAR-10.** Better balancing can be accomplished through the adjustments of the balancing parameters.

**Network architecture**  The encoder consists of a backbone followed by a projector. We employ a variant of ResNet-18 for CIFAR-10 as the backbone and a two-layered fully-connected MLP as the projector. For the projector, the input and output dimensions of the first layer and 512 and 2,048, respectively, and the input and output dimensions of the second layer are 2,048. Batch normalization is applied to all layers, and the ReLU activation function is applied to the first layer.

**Pretraining configuration**  We pretrain the encoder with 512 batch size for 200 epochs. We employ the SGD optimizer and set the momentum to 0.9, the learning rate to 0.1, and the weight decay rate to 0.0001.

**Evaluation configuration**  We train the linear classifier with a batch size of 256 for 90 epochs using SGD with momentum 0.9 and learning rate 30, and apply a cosine decay schedule.

### A.4.2   Standard evaluations

Table 4 presents a set of standard evaluations. Error bars, represented as the mean $\pm$ standard deviation, are reported based on five independent runs. We choose $(\alpha, \lambda)$ as $(4, 2)$ and $(2, 4)$ for ImageNet and CIFAR-10, respectively. We also include $k$-nearest neighbors evaluation. Specifically, we retrieve the $k$ nearest training image representations for a given test image representation. Their respective labels are aggregated using a majority voting process to predict the label for the test image. In ImageNet experiments, $k$ is set to 200, whereas in CIFAR-10 experiments, $k$ is set to 1.

### A.4.3   Impact of balancing parameters on CIFAR-10

As in Section 6, Figure 5 shows that, balancing between the attracting component and the repelling component is important using balancing parameters $\alpha$ and $\lambda$.

Table 5: Comparison of class distributions under balanced contrastive loss. The results show that the uniform class distribution leads to better performance.

| | Class distribution | |
| --- | --- | --- |
| | Uniform | Long-tailed |
| | 21.24 | 15.01 |

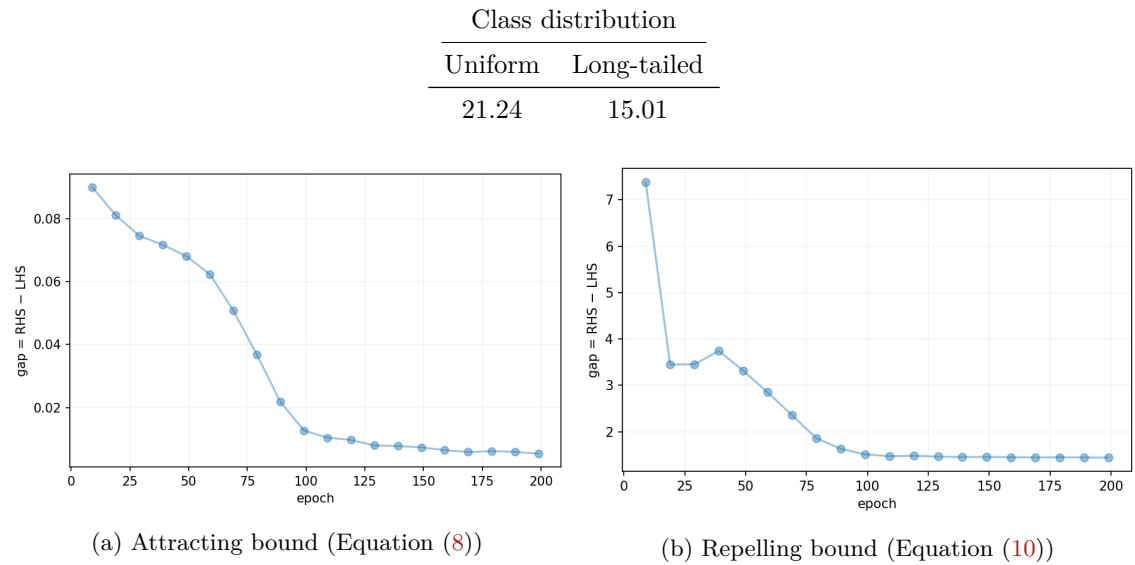

(a) Attracting bound (Equation (8))

(b) Repelling bound (Equation (10))

Figure 6: **Bound tightness**: mean gap $\Delta = \text{RHS} - \text{LHS}$ over training checkpoints. Lower is tighter.

### A.4.4  Impact of data imbalance on the balanced contrastive loss

As an extension of Section 5.4, we investigate the impact of data imbalance on the balanced contrastive loss in Table 5. We adopt the balancing parameters $\alpha = 2$ and $\lambda = 1$ for comparison, as the SimCLR loss is equivalent to the generalized NT-Xent loss under this setting. Compared to SimCLR, the balanced contrastive loss exhibits relatively improved performance. Nevertheless, similar to SimCLR, performance is higher when the class distribution is balanced. This observation aligns well with our theoretical framework, which assumes uniformity in class distribution.

### A.4.5  Tightness of the upper bounds

We quantify the tightness of our upper bounds in Equation (8) and Equation (10) by measuring the per-sample gap $\Delta = \text{RHS} - \text{LHS}$ and reporting the mean across the dataset. We train on CIFAR-10 and store checkpoints every 10 epochs. For each checkpoint we evaluate: (i) the attracting bound using $K = 10$ Monte Carlo samples of $T$ to approximate $\mathbb{E}_T$; (ii) the repelling bound using $K = 1$ draw of $T'$ and a memory bank of $M = 50{,}000$ negatives (all training images) to approximate $\mathbb{E}_{T'}$ and $\mathbb{E}_{X'}$, respectively.

Figure 6 shows the epoch-wise mean gap for the attracting and repelling components, respectively. Both gaps decrease and then stabilize at a small value, indicating that the bounds become tight as training progresses. The repelling bound shows a consistently larger mean gap than the attracting bound across epochs.

**Equality conditions.** For the attracting bound, the proof of Equation (8) has slack only from Jensen's inequality on the norm: $\|\mathbb{E}_T f_\theta(T(x))\| \leq \mathbb{E}_T \|f_\theta(T(x))\| = 1$. Hence, the equality holds when $\|\mathbb{E}_T f_\theta(T(x))\| = 1$, i.e., all augmented views of the same image map to the same unit vector (view-invariance).

For the repelling bound, the proof of Equation (10) uses several inequalities. In practice, tightness is approached when similarities $s(f_\theta(t(x)), f_\theta(T'(X')))$ vary little across $T'$ and $X'$, so moving expectations through exp and log adds negligible slack. It is also approached when a single negative class dominates or $\alpha$ is large, making $\frac{1}{\alpha} \log \sum \exp(\alpha \cdot) \approx \max$.

