# OpenReview forum: "Understanding Self-supervised Contrastive Learning through Supervised Objectives"
_TMLR — Accepted by TMLR_

### Review · Reviewer_eNUj · 2025-06-26

**Summary Of Contributions:**

The authors attempt to provide a novel formulation of self-supervised InfoNCE loss from the lens of supervised learning. The authors first derive the theoretical foundations of the supervised contrastive loss and then move on to the SSL counterpart.

**Audience:**

No

**Claims And Evidence:**

No

**Requested Changes:**

1. The authors should provide a comparison with contemporary baselines on benchmark datasets like CIFAR10, CIFAR100, and ImageNet100. Otherwise, it would be difficult to decide the merit of the proposed method based on the results of just the proposed method.

2. The authors should clarify how the authors are constructing the negative pairs or prototypes for the targets in more detail. The current discussion does not provide enough explanation.

3. The authors should also discuss how their proposed method is different from the previous baselines, and how it improves the performance in terms of metrics like uniformity, alignment, etc.

**Strengths And Weaknesses:**

**Strengths:**

1. The authors give a theoretical representation of the supervised contrastive loss as an optimisation problem using a concept of prototype representation.

2. Breaks down supervised contrastive loss in terms of attraction loss and repulsion loss, which is intuitive.

**Weakness:**

1. Assumption 4.1 can be placed before mentioning $s(.,.)$ : the similarity measure.

2. *Construction of Surrogate Prototype*: On one hand, the authors do not have access to labels, and on the other hand the authors state that they construct the surrogate prototype for the target image for $t(x)$ as the expectation of the representations of the available images sharing the same label as $t(x)$.
It is unclear how the authors calculate the prototypes of the different classes without any pseudo-labelling step. There are several approaches which use pseudo-labelling for assigning labels for supervisory information, and also handle the attract and repel components differently [1].

3. Again in Eqn. (7), how do the authors differentiate between different labels $y'$ and $y$, if the authors do not have any access to the labels? The process needs to be discussed in more detail.

4. The expression for Eqn. (13) is similar to DCL [2], as it does not contain the positive pair in the denominator. The authors should compare the performance with the contemporary SSL methods to establish the effectiveness of the proposed method.

5.  The approach proposed is also very similar to the approach taken in Wang and Isola (2020) [3] and MIO [4], in that the authors divide the original objective into two terms, alignment (attract) and uniformity (repel). How is the proposed method different from the existing methods, and what improvements does it bring in terms of performance or convergence?

6. The authors mention SimCLR but do not compare it in their manuscript. There is no comparison present in the whole paper. It is advisable to compare it with the latest methods like [5].

7. The authors also present results on long-tailed experiments, but again, no comparison is present with methods like MACL [1], DySTreSS [6], Kukleva et al. (2023) [7], etc.

References:

[1] Huang, Z., Chen, H., Wen, Z., Zhang, C., Li, H., Wang, B., and Chen, C., 2023. Model-aware contrastive learning: towards escaping the dilemmas. In Proceedings of the 40th International Conference on Machine Learning (ICML'23), Vol. 202. JMLR.org, Article 559, 13774–13790.

[2] Yeh, CH., Hong, CY., Hsu, YC., Liu, TL., Chen, Y., LeCun, Y. (2022). Decoupled Contrastive Learning. In: Avidan, S., Brostow, G., Cissé, M., Farinella, G.M., Hassner, T. (eds) Computer Vision – ECCV 2022. ECCV 2022.

[3] T. Wang, and P. Isola, 2020. Understanding contrastive representation learning through alignment and uniformity on the hypersphere. In Proceedings of the 37th International Conference on Machine Learning (ICML'20), Vol. 119. JMLR.org, Article 921, 9929–9939.

[4] S. Manna, U. Pal, S. Bhattacharya, MIO: Mutual Information Optimization using Self-Supervised Binary Contrastive Learning, abs/2111.12664

[5] Y. Zhang, H. Zhu, Z. Song, Y. Chen, X. Fu, Z. Meng, P. Koniusz, and I. King, “Geometric view of soft decorrelation in self-supervised learning,” in Proceedings of the 30th ACM SIGKDD Conference on Knowledge Discovery and Data Mining, pp. 4338–4349, 2024

[6] S. Manna, S. Chattopadhyay, R. Dey, U. Pal and S. Bhattacharya, "Dynamically Scaled Temperature in Self-Supervised Contrastive Learning," in IEEE Transactions on Artificial Intelligence, vol. 6, no. 6, pp. 1502-1512, June 2025, doi: 10.1109/TAI.2024.3524979.

[7] A. Kukleva, M. B ¨ohle, B. Schiele, H. Kuehne, and C. Rupprecht, “Temperature schedules for self-supervised contrastive methods on long-tail data,” in The Eleventh International Conference on Learning Representations, 2023.

---

> ### Author Response · Authors · 2025-08-13
>
> We sincerely appreciate your valuable time and effort in reviewing our manuscript. We respond to each comment below.
>
> ---
> ### **[W1] Placement of Assumption 4.1**
>
> We intentionally present the problem formulation in Section 3 in a general form, introducing specific assumptions in Section 4 to guide the theoretical derivation. To address the reviewer’s concern, we have added an example (“e.g., cosine similarity”) when the similarity measure first appears in Section 3.
>
> ---
> ### **[W2] Construction of Surrogate Prototype**
>
> To clarify, the surrogate prototype representation is not the expectation over all images sharing the same label. Rather, it is the expectation over augmented views of the same image $x$: $\\tilde{\\mu}\_{y} := \\mathbb{E}\_{T}f\_{\\theta}(T(x))$. We realize that the phrase “available images sharing the same label” may have caused confusion. What we intended to convey is that augmented views are images that share the same (unobserved) label as the original image. We have revised Section 3.2 to make this point clearer.
>
>
>
> ---
> ### **[W3] On Equation (7)**
>
> The repelling component in Equation (7) presents the ideal objective. Although it appears to require label information, under the stated assumptions it can be bounded by a term that does not require any labels (see the derivation following Equation (42)). For this reason, we stated immediately after Equation (7) that “we leave the repelling component as is since it can be managed without modification.”
>
> ---
> ### **[W4] Relation to Decoupled Contrastive Loss (DCL)**
>
> Indeed, Equation (13) shares the decoupled denominator structure with DCL, and we explicitly discuss this connection in Section 5.1. Thus, our framework provides a theoretical motivation for this form, as it naturally arises from approximating a supervised objective. As such, the balanced contrastive loss reduces to DCL when $\\lambda=1$, allowing direct comparison in Figure 4(a).
>
> ---
> ### **[W5] Comparison with [3] and [4]**
>
> Many studies on contrastive learning deal with losses consisting of an attracting term and a repelling term, so they may appear similar. Below, we outline the key differences.
>
> In Section 2, we address [3]. [3] and our paper differ in terms of approach, objective, and theoretical setting.
>
> * Approach: [3] takes the established contrastive loss (Equation (1) in [3]) as given. In contrast, our paper begins with a formulated supervised objective from which we derive the contrastive loss.
>
> * Objective: [3] explores how attracting and repelling other samples relates to alignment and uniformity properties. In contrast, our paper suggests how attracting and repelling pseudo-labels can result in attracting and repelling other samples.
>
> * Theoretical setting: [3] primarily examines the limiting behavior in an asymptotic setting (Theorem 1 in [3]) where the number of negative samples approaches infinity. On the other hand, our approach addresses a more general setting.
>
> [4] also starts from an InfoNCE-type loss constructed with samples and then applies several modifications, such as removing the positive–positive repulsion term and replacing the negative term with its exponential upper bound. In contrast, our work starts from a supervised objective constructed with pseudo-labels (prototype representations) and, under approximation, derives an InfoNCE-type loss as a consequence.
>
>
> ---
> ### **[W6] Comparison with SimCLR**
>
> We would like to clarify that our method reduces to SimCLR when $\\lambda=1$ in the generalized NT-Xent loss (Figure 4(b)). Please also note that our work is primarily aimed at understanding contrastive losses (theoretical perspective and insights) through the lens of supervised objectives rather than empirical benchmarking.
>
> ---
> ### **[W7] Purpose of long-tailed distribution experiments**
>
> The results on long-tailed distributions in our paper are intended to empirically illustrate the importance of Assumption 4.5 (balanced dataset) in our theoretical framework, rather than to propose a method specifically designed for long-tailed recognition or to compete with specialized methods.
>
> ---
> ### **[R1] On benchmark datasets**
>
> We would like to clarify that experimental results on CIFAR-10 are provided in Appendix (Table 4 and Figure 5). These results are consistent with our theoretical insights, showing similar trends on CIFAR-10 as on ImageNet.
>
> ---
> ### **[R2] Construction of negative pairs and prototypes**
>
> As for prototype construction, please refer to our response in [W2]. Regarding negative pairs, we have provided a detailed explanation at the beginning of Section 6.
>
> ---
> ### **[R3] On previous baselines**
>
> Please refer to our responses in [W4], [W5], and [W6] for detailed discussions on how our method differs from previous baselines.

---

### Review · Reviewer_jnBJ · 2025-07-02

**Summary Of Contributions:**

This paper presents a theoretical framework that aims to bridge supervised prototype-based representation learning and self-supervised contrastive learning. The authors start from a supervised formulation in which image representations are attracted to class-specific prototype vectors and repelled from those of other classes. This objective is then approximated in a self-supervised setting by replacing ground-truth prototypes with surrogate representations constructed via data augmentations. Under common assumptions such as cosine similarity and normalized embeddings, the authors derive tractable upper bounds for both the attractive and repulsive components of the loss, resulting in a contrastive loss that resembles the InfoNCE loss used in SimCLR.

Building on this formulation, the paper proposes a balanced contrastive loss that decouples the attractive and repulsive terms and introduces two tunable hyperparameters controlling their relative influence. Empirical results on ImageNet with linear evaluation show that certain settings of these parameters can outperform the standard NT-Xent loss. The paper also discusses how factors such as the choice of similarity measure, data augmentation strategy, dataset balance, and network architecture relate to the theoretical assumptions in the proposed framework.

**Audience:**

Yes

**Claims And Evidence:**

Yes

**Requested Changes:**

Overall, the paper is technically sound and well written — the derivations are clear, and the ideas are easy to follow.  I found myself questioning the necessity of the work at a more fundamental level.  The core perspective is not surprising and has been implicitly or explicitly acknowledged in quite a few prior works.  While this paper formalizes that intuition nicely, it's not clear to me what new understanding or actionable insight this framing really provides.  I would encourage the authors to more directly address what makes this perspective worth formalizing now, how it meaningfully differs from existing interpretations, and who it is intended to benefit.

**Strengths And Weaknesses:**

Strengths:

The paper is clearly written and well-organized. The derivations are presented in a step-by-step manner, which makes the theoretical ideas accessible to a wide range of readers. The core idea — to interpret contrastive self-supervised learning as an approximation to prototype-based supervised learning — can help unify existing intuitions under a formal framework. The empirical study, though limited in scope, is cleanly executed and provides useful insights into how different hyperparameter settings affect performance.

Weaknesses:

The conceptual link between contrastive learning and prototype clustering has been widely acknowledged in the literature, and this work mainly formalizes that intuition without introducing fundamentally new theoretical results. The narrative, interpreting self-supervised learning as approximating supervised objectives, is already prevalent in prior methods. The proposed loss function is effectively a hyperparameterized variant of NT-Xent. Similar formulations—with tunable temperatures or weighting between positive and negative terms—have already been explored in prior work. The improvement over NT-Xent is achieved primarily through grid search over \alpha and \lamda, which raises questions about practical significance.

Overall, the paper functions more as an explanatory or interpretive work rather than a contribution that is likely to shift future research directions or practice.

---

> ### Author Response · Authors · 2025-08-13
>
> ---
> ### **[W1, R1] On contributions**
>
> We acknowledge that the conceptual link between contrastive learning and prototype clustering has been noted in prior work. Our contribution lies in providing a formal derivation of this link from a supervised objective, showing under what approximations and assumptions the commonly used contrastive form emerges, and unifying existing design heuristics within this theoretical framework. Although the resulting loss bears a resemblance to NT-Xent, it more precisely matches the structure of DCL (Decoupled Contrastive Loss), as our derivation naturally leads to a decoupled denominator form.
>
> The primary goal of our work is to advance the theoretical understanding of contrastive learning rather than to introduce a new SSL method. Although the high-level intuition is familiar, a clear and general derivation remains underexplored in the literature. We provide a citable reference point. In particular, we formalize prototype representation bias and a balanced contrastive loss, offering interpretable guidance for method design and evaluation.

---

> > ### Author Response · Authors · 2025-08-13
> >
> > We sincerely appreciate your valuable time and effort in reviewing our manuscript.

---

### Review · Reviewer_T56E · 2025-08-06

**Summary Of Contributions:**

The paper proposes a theoretical framework to study self-supervised representation learning as an approximation to supervised representation learning. The core contribution lies in demonstrating how popular contrastive losses, particularly those resembling InfoNCE, can be derived from a supervised objective function. They do so by employing the concept of prototype representation of a visual concept (in the case of images). The prototype representation serves as a target for images with the same visual concept. The key idea is that for SSL, where labels are not available, a surrogate prototype representation can be used as the target. This surrogate prototype representation is the expectation, over the augmentations, of the representations of the available images.

A significant new concept introduced is the "prototype representation bias", defined as the discrepancy between the ideal supervised prototype representation and its self-supervised surrogate. The authors empirically demonstrate that reducing this bias correlates with improved downstream performance, offering a new metric and target for SSL research.

The paper provides several theoretical justification for common practices, such as representation normalization, cosine similarity,  and balanced class distribution (assumptions of the theorems and empirically validated); the siamese network architecture that naturally aligns with the structure of the attractive component; While I cannot comment on the absolute novelty of the core ideas due to my lack of familiarity with this specific topic, I find the paper convincing and recommend acceptance.

**Audience:**

Yes

**Broader Impact Concerns:**

I don't have any concerns on the ethical implications of this work.

**Claims And Evidence:**

Yes

**Requested Changes:**

The authors could revise the phrasing in Section 3.2 to avoid potential confusion regarding the term "sharing the same label." A more precise and explicit definition of the self-supervised signal, such as "sharing the same pseudo-label based on data augmentation," would improve clarity.

While the theoretical derivation of the balanced contrastive loss is a key contribution, the paper should more explicitly position its practical novelty.

The paper should include a discussion about the tightness of the bounds derived in the proofs, particularly for the repelling component. The authors could add a brief comment in the main text or the appendix acknowledging that the long chain of inequalities, while crucial for motivating the loss's structure, may result in a loose bound. Perhaps emphasizing that the bound's structural properties are most important to the argument.

**Strengths And Weaknesses:**

In Section 3.2, the paper defines the surrogate prototype representation using "images sharing the same label as t(x)." While the explanation clarifies this refers to augmented views of the same original image, the phrasing could initially be confusing for a self-supervised context where explicit labels are absent.

The proposed balanced contrastive loss is presented as a "natural extension" of InfoNCE and is derived from Equation (12). While its theoretical motivation is clear, its practical novelty compared to existing variants of contrastive losses (e.g., decoupled contrastive loss mentioned in Section 5.1) might be incremental. The primary contribution here seems to be the theoretical framing and the empirical demonstration of the importance of balancing parameters, rather than a fundamentally new loss formulation.

As far as I can tell the two main proofs and related lemmas are correct. My concern is that the chain of inequalities, although instrumental in connecting the theoretical model of hard negative mining (repelling the single closest negative prototype) to a tractable, sample-based loss function (repelling multiple negative samples), likely results in a loose bound. For example, a high variance in the distribution of f(T(x)) would likely lead to a looser bound.

---

> ### Author Response · Authors · 2025-08-13
>
> We sincerely appreciate your valuable time and effort in reviewing our manuscript. We respond to each comment below.
>
> ---
> ### **[W1] Surrogate prototype representation definition**
> Thank you for pointing this out. We revised the description in Section 3.2 as follows to avoid any confusion:
>
> > We construct it as the expectation of the representations of augmented views of the image $x$, i.e., $\\tilde{\\mu}\_{y} := \\mathbb{E}\_{T}f\_{\\theta}(T(x))$.
> >
> > Since data augmentation preserves labels, augmented views share the same (unobserved) label $y$.
>
> ---
> ### **[W2] The primary contribution here seems to be the theoretical framing**
>
> We agree that the theoretical framing of the balanced contrastive loss is the primary contribution of our work. From a practical standpoint, our aim is to help understand the roles of the balancing parameters in our framework and to empirically report how varying them affects performance. We have incorporated this statement at the beginning of Section 6 in the revised manuscript.
>
> ---
> ### **[W3] Discussion on the tightness of the derived bounds**
>
> We added Section A.4.5 and Figure 6 quantifying tightness via $\\Delta=\\mathrm{RHS}-\\mathrm{LHS}$. In our experiments, the repelling bound exhibits a larger gap than the attracting bound.

---

### Decision · Action_Editor_3eK3 · 2025-10-07

**Recommendation:** Accept with minor revision

**Additional Comments:**

This paper attempts to investigate how each component of self-supervised learning affects the whole learning process. Whereas the provided insights are more or less rediscoveries in prior work (sometimes only implied implicitly), the authors' analysis is still valuable to demonstrate them objectively. Most of the reviewers agreed on this point.

While some reviewers do not see significant practical benefits of the proposed loss function, this is acceptable because the proposed loss function does not aim to achieve the state-of-the-art performance but for the analysis purpose in AE's take. Thus, AE does not overly underestimate this aspect.

Upon the acceptance, we expect the authors to address the remaining points discussed between the authors and reviewers. Specifically, the literature comparison (raised by Reviwer eNUj) can be further expanded in the revision.
Additionally, I think the subscript $y$ in the definition of surrogate prototype representation (in eq. (6)) is strange because the representation should not depend on $y$ at all in this case. Please consider correcting the notation appropriately.

**Audience:**

Yes

**Audience Explanation:**

The mechanism of self-supervised learning has been still unclear and an important question opened to our community, which is partially addressed as the contributions of this paper.

**Claims And Evidence:**

Yes

**Claims Explanation:**

The authors conduced a careful analysis to components consisting self-supervised learning, such as the loss function, bias induced by data augmentation, similarity measure, dataset skewness, and architecture, to demonstrate their influence on learning performance. The results are delicately presented and appropriate for providing evidence of their claims.